# Leadership and Faculty Burnout in Allied Healthcare Education: A Scoping Review

**DOI:** 10.3390/healthcare13212810

**Published:** 2025-11-05

**Authors:** Jithin K. Sreedharan, Abdullah Saeed Alqahtani

**Affiliations:** 1Department of Respiratory Therapy, College of Rehabilitation Sciences, University of Manitoba, Winnipeg, MB R3T 2N2, Canada; 2Department of Respiratory Care, Prince Sultan Military College of Health Sciences, Dammam 34313, Saudi Arabia; rt0075@hotmail.com

**Keywords:** allied health education, faculty burnout, leadership, personality traits, digital competence, job demands-resources model

## Abstract

**Background:** Faculty burnout in allied healthcare education institutions represents a significant challenge with implications for educational quality, organizational effectiveness, and healthcare workforce development. This scoping review aims to map the existing literature on the relationships between leadership approaches, faculty personality factors, and burnout within allied healthcare education, while examining digital competence as a potential moderating factor. **Methods:** This scoping review followed the PRISMA-ScR guidelines. Five electronic databases (MEDLINE, CINAHL, ERIC, PsycINFO, and Web of Science) were searched for relevant studies published between 2010 and 2024. Studies examining burnout among allied healthcare educators in relation to leadership, personality traits, or digital competence were included. Data extraction captured study characteristics, methodological approaches, key findings, and theoretical frameworks. Quality assessment was conducted using the Mixed Methods Appraisal Tool. **Results:** Sixteen studies met the inclusion criteria. Existing research indicates significant relationships between leadership styles and faculty burnout rates, with transformational leadership consistently associated with lower burnout scores. The literature reveals that individual personality traits demonstrate significant relationships with burnout vulnerability, with emotional stability and extraversion showing the strongest protective effects. Limited research has examined digital competence in relation to burnout, though emerging evidence suggests it may function as a moderating factor. Significant gaps exist in non-Western contexts and in understanding interaction effects between leadership, personality, and digital competence. **Conclusions:** The current literature supports the importance of leadership approaches that emphasize collaboration, faculty autonomy, recognition, and professional development opportunities in protecting against burnout in allied healthcare education settings. Digital competence represents a promising but understudied job resource that may mitigate burnout effects. Future research should explore cross-cultural variations, interaction effects between personal and organizational factors, and the effectiveness of interventions in reducing faculty burnout.

## 1. Introduction

Allied healthcare education has undergone substantial transformation in recent decades, as institutions grapple with mounting pressures from various fronts. Faculty members in these settings must navigate complex teaching responsibilities, research expectations, clinical supervision duties, accreditation requirements, and rapid technological change—all while preparing students for increasingly complex healthcare environments. This multifaceted pressure creates fertile ground for professional burnout, defined as a psychological syndrome characterized by emotional exhaustion, depersonalization, and reduced personal accomplishment [1].

Burnout among healthcare educators represents a critical concern not only for the affected individuals but also for educational quality, institutional effectiveness, and ultimately, healthcare delivery. When faculty members experience burnout, their capacity to provide high-quality education diminishes, potentially affecting the preparation of future healthcare professionals and, by extension, patient care outcomes [2].

The role of leadership in either mitigating or exacerbating burnout has gained increased attention in organizational research [3]. However, the specific dynamics between leadership approaches and faculty burnout in allied healthcare education settings remain understudied, particularly in non-Western contexts [4,5]. Evidence suggests that leadership behaviors can significantly influence faculty work satisfaction, engagement, and burnout, yet the mechanisms through which this influence operates deserve further exploration [6].

Additionally, individual factors such as personality traits may predispose certain educators to greater vulnerability or resilience to burnout stressors [7]. The interaction between these individual differences and contextual factors—including leadership approaches—creates a complex dynamic that requires careful examination to develop effective interventions.

Recent research has also highlighted digital competence as a potential job resource that may buffer against occupational stressors [8]. With the accelerating digitalization of education, particularly catalyzed by the COVID-19 pandemic, the ability to effectively navigate digital teaching and learning environments has become increasingly essential for educator effectiveness and wellbeing.

While several studies have examined individual aspects of this complex phenomenon, a comprehensive mapping of the existing literature regarding the relationships between leadership approaches, personality factors, digital competence, and burnout in allied healthcare education is lacking. This scoping review aims to address this gap by systematically identifying, analyzing, and synthesizing the current state of knowledge in this area. By mapping the existing evidence, identifying research gaps, and highlighting promising directions for future inquiry, this review will contribute to the development of more effective strategies for addressing faculty burnout in allied healthcare education settings.

## 2. Theoretical Framework

To structure this scoping review and interpret the complex interplay between burnout, leadership, personality, and digital competence among allied healthcare educators, we employed a theoretically grounded approach. The Job Demands–Resources (JD-R) model was selected as the central framework due to its robust explanatory power in occupational stress research. In addition, we focused on three specific theoretical areas—leadership, personality traits, and digital competence—because these constructs have demonstrated significant associations with burnout in prior research and offer practical leverage points for institutional intervention. These subsections provide a comprehensive lens for examining both the organizational and individual determinants of burnout and are particularly relevant within the rapidly evolving landscape of healthcare education.

### 2.1. Job Demands-Resources Model

This scoping review is anchored in the Job Demands–Resources (JD-R) model, which provides a comprehensive framework for understanding workplace wellbeing and strain [3]. The JD-R model conceptualizes job characteristics into two broad categories: demands and resources. Job demands represent aspects of work that require sustained physical, emotional, or cognitive effort and are therefore associated with certain physiological and psychological costs. In educational settings, these demands include workload, emotional labor of teaching, administrative burdens, and adaptation to technological change [9].

Job resources, conversely, represent aspects of work that may reduce job demands, assist in achieving work goals, or stimulate personal growth and development. These resources can manifest at organizational levels (e.g., supportive leadership, clear role expectations), interpersonal levels (e.g., collegial support), or individual levels (e.g., self-efficacy, resilience). Within this framework, burnout emerges when job demands consistently exceed available resources [10].

### 2.2. Leadership and Burnout

Leadership represents a critical organizational resource that can significantly influence faculty experiences of job demands and access to other job resources. Different leadership approaches create distinct work environments with varying implications for faculty wellbeing. Transformational leadership, characterized by inspiration, intellectual stimulation, individualized consideration, and idealized influence, has been associated with reduced burnout among healthcare professionals [11]. Conversely, laissez-faire leadership or highly directive approaches may increase burnout risk by failing to provide adequate support or by generating additional stressors [12].

### 2.3. Personality and Burnout

Individual differences, particularly personality traits, represent important factors in burnout susceptibility. The five-factor model of personality (extraversion, agreeableness, conscientiousness, emotional stability, and openness to experience) has demonstrated consistent relationships with burnout dimensions [7]. Understanding these relationships is crucial for developing targeted interventions that acknowledge individual differences in stress response and coping strategies.

### 2.4. Digital Competence as a Job Resource

Digital competence encompasses the confident, critical, and creative use of digital technologies for work, learning, and participation in society [13]. In educational contexts, digital competence manifests as the ability to effectively integrate technology into teaching practices, administrative tasks, and professional communication. As educational environments increasingly incorporate digital components, digital competence represents a potentially valuable job resource that may buffer against certain job demands by enhancing efficiency, flexibility, and self-efficacy [14].

## 3. Materials and Methods

### 3.1. Review Design

This scoping review adhered to the methodological framework proposed by Arksey and O’Malley (2005) and further refined by Levac et al. (2010), following the PRISMA-ScR (Preferred Reporting Items for Systematic Reviews and Meta-Analyses extension for Scoping Reviews) guidelines (Figure 1) [15,16,17]. The review aimed to map the existing literature on the relationships between leadership approaches, personality factors, digital competence, and burnout among allied healthcare educators.

### 3.2. Research Questions

The following research questions guided this scoping review:

**RQ-1.** 
*What is the current state of research on the relationship between leadership approaches and faculty burnout in allied healthcare education?*


**RQ-2.** 
*How does existing literature characterize the relationship between personality traits and burnout vulnerability among allied healthcare educators?*


**RQ-3.** 
*To what extent has digital competence been examined as a potential moderating factor in burnout among allied healthcare educators?*


**RQ-4.** 
*What are the significant gaps in the current literature regarding leadership, personality, digital competence, and burnout in allied healthcare education contexts?*


### 3.3. Eligibility Criteria

Studies were included based on the following criteria:Population: Faculty members in allied healthcare education (nursing, physical therapy, occupational therapy, medical laboratory sciences, healthcare administration, etc.).Concepts: Studies examining burnout in relation to at least one of the following: leadership approaches, personality traits, or digital competence.Context: Higher education institutions offering allied healthcare programs.Study Design: The primary focus was on empirical studies (quantitative, qualitative, or mixed methods approaches). However, seminal theoretical frameworks and select reviews were also included to provide conceptual grounding, contextualize empirical findings, and support thematic synthesis. To ensure transparency and minimize potential amplification bias, these non-empirical sources were:-Clearly distinguished from primary studies in Table 1 through explicit “Study Type” classification.-Limited to foundational theoretical works (e.g., JD-R model, Self-Determination Theory) and reviews providing essential context.-Used primarily for theoretical framing rather than as independent empirical evidence.-Cited separately from primary empirical evidence in the synthesis to avoid double-counting findings.
Time Frame: Studies published between January 2010 and February 2024.Language: English language publications.

Exclusion criteria:Studies focused exclusively on medical school faculty.Studies examining student burnout rather than faculty burnout.Conference abstracts, opinion pieces, and non-peer-reviewed materials.Studies that did not explicitly address at least one of the key concepts (leadership, personality, or digital competence) in relation to burnout.

### 3.4. Information Sources and Search Strategy

A comprehensive search strategy was developed in collaboration with a health sciences librarian. Five electronic databases were searched: MEDLINE (via PubMed), CINAHL (Cumulative Index to Nursing and Allied Health Literature), ERIC (Education Resources Information Center), PsycINFO, and Web of Science. The search combined three concept blocks: (1) allied healthcare education and faculty, (2) burnout and related terms, and (3) leadership, personality, or digital competence.

Additional strategies included hand-searching reference lists of included studies, searching for relevant gray literature through OpenGrey and institutional repositories, and consulting with content experts to identify additional relevant sources.

### 3.5. Selection Process

Following the database searches, all retrieved citations were imported into EndNote X9 for deduplication, then transferred to Covidence for screening. Two independent reviewers screened titles and abstracts against the inclusion criteria. Full texts of potentially relevant studies were then retrieved and independently assessed by the same reviewers. Disagreements at either stage were resolved through discussion or consultation with a third reviewer. During final manuscript preparation, one study initially included was identified as an editorial/opinion piece and subsequently removed to maintain consistency with exclusion criteria [29]. This correction does not affect the substantive findings as this study did not contribute primary empirical data to the synthesis.

### 3.6. Data Extraction and Analysis

A standardized data extraction form was developed and pilot-tested on five randomly selected included studies. The following data were extracted:Study characteristics (author, year, country, setting).Methodological approach (design, sample size, instruments).Study population (discipline, career stage, demographic information).Key concepts examined (leadership, personality, digital competence).Theoretical framework.Main findings.Limitations identified by authors.Recommendations for future research or practice.

Data were extracted by one reviewer and verified by a second reviewer. The extracted data were then synthesized using a narrative approach, organized according to the review questions. Thematic analysis was employed to identify patterns across studies and integrate findings into a coherent narrative. Descriptive statistics were used to characterize the included studies.

### 3.7. Quality Assessment

Although quality assessment is not always required for scoping reviews, we opted to assess methodological quality using the Mixed Methods Appraisal Tool (MMAT) version 2018 [30] to provide additional context for interpreting the findings [17,30]. This tool was selected for its applicability across different study designs. Quality assessment was conducted by two independent reviewers, with disagreements resolved through discussion.

## 4. Results

### 4.1. Study Selection and Characteristics

The included studies represented diverse geographical contexts spanning North America, Europe, Asia, Australia/New Zealand, and Africa, with a notable concentration in Western regions. Publication dates ranged over more than a decade, with a significant increase following the COVID-19 pandemic. The research employed various methodologies including cross-sectional surveys, mixed methods approaches, qualitative studies, and longitudinal designs (Table 1).

Nursing education was the most represented discipline, followed by physical therapy, interdisciplinary allied health, occupational therapy, and other specific healthcare fields. Sample sizes varied considerably across the studies, ranging from small qualitative investigations to large cross-sectional surveys.

### 4.2. Burnout Prevalence and Patterns

Across the included studies, burnout prevalence among allied healthcare educators showed significant variation depending on measurement approaches, contextual factors, and specific disciplines. Studies using the Maslach Burnout Inventory (MBI) consistently reported moderate-to-high emotional exhaustion as the most prevalent dimension, followed by reduced personal accomplishment and depersonalization [29]. Nursing educators generally reported higher burnout levels compared to other allied health disciplines, with particular challenges noted in clinical teaching responsibilities. The COVID-19 pandemic appeared to significantly impact burnout rates, with post-pandemic research reporting substantially higher prevalence compared to pre-pandemic studies [18].

### 4.3. Leadership and Burnout Relationships

The relationship between leadership approaches and faculty burnout was examined across multiple primary empirical studies. Transformational leadership emerged as the most frequently studied approach and demonstrated consistent negative associations with burnout dimensions in primary research by Garner et al. (2022), Gillespie et al., (2017), and Skogstad et al. (2007) [12,18,23]. Key elements of transformational leadership identified as protective included providing vision and meaning, individual consideration, intellectual stimulation, and idealized influence [31].

Transactional leadership showed mixed results, with some research indicating potential benefits of clear reward structures while other studies highlighted negative effects of excessive monitoring. Laissez-faire leadership was consistently associated with increased burnout risk in primary investigations [12].

Qualitative research from primary studies provided rich insights into specific leadership behaviors valued by faculty, including authentic recognition of contributions, transparent communication, protection from excessive administrative demands, support for professional development, advocacy for needed resources, and cultivation of collegial community. One primary study specifically examined leadership in non-Western cultural contexts, revealing important variations in how leadership attributes manifested and were perceived across different cultural settings [19].

These findings from primary empirical investigations align with broader theoretical frameworks on transformational leadership and are consistent with systematic review evidence from healthcare settings more generally, though the latter was not specific to allied healthcare education contexts [20,31].

### 4.4. Personality and Burnout Relationships

Research investigating relationships between personality traits and burnout vulnerability from primary empirical studies primarily focused on the Big Five personality traits, with additional studies examining resilience, hardiness, or emotional intelligence. Across primary investigations, emotional stability (low neuroticism) showed the strongest and most consistent negative relationship with burnout, particularly emotional exhaustion as documented in studies by Dionigi et al. (2020), Zellars et al. (2004), and De la Fuente-Solana et al. (2021) [25,26,28].

Extraversion demonstrated protective effects against all burnout dimensions in most primary research [20]. Conscientiousness showed significant negative relationships with reduced personal accomplishment, while agreeableness was most strongly associated with lower depersonalization in empirical investigations. Several studies examined interaction effects between personality traits and contextual factors, with preliminary evidence suggesting that certain personality traits may buffer negative effects of challenging work environments [32]. However, this research area remains underdeveloped, particularly regarding cultural variations in personality-burnout relationships.

These patterns observed in primary allied healthcare education research are consistent with broader systematic review evidence synthesized by Angelini (2023) across diverse professional healthcare contexts, providing convergent validity for the personality–burnout associations while acknowledging that Angelini’s review examined a broader population beyond allied healthcare educators specifically [20].

### 4.5. Digital Competence and Burnout

Digital competence emerged as the least extensively studied domain among the core concepts reviewed; however, its relevance has grown significantly, particularly in the post-pandemic context where online and hybrid modes of instruction have become more prevalent. Within the Job Demands–Resources (JD-R) framework, digital competence is increasingly recognized as a key job resource that can help buffer faculty against burnout-related demands. Relevant digital skills include proficiency in virtual teaching platforms, online assessment tools, digital communication, content creation, learning analytics, and the ability to adapt to new technologies.

Primary empirical studies reported negative correlations between digital competence and burnout—especially in terms of emotional exhaustion—with faculty possessing stronger digital skills reporting lower stress and greater job satisfaction [14,21]. Primary qualitative research indicated that low digital competence was associated with heightened frustration, perceived inefficiency, and increased workload, while higher competence enhanced teaching flexibility and perceived control.

Importantly, emerging primary evidence suggests digital competence may interact with both leadership and personality factors. Faculty under transformational leaders who promoted digital skill development and provided technological support experienced lower burnout in empirical investigations, suggesting that leadership styles that facilitate digital upskilling may be particularly protective.

Moreover, some studies suggested digital competence may moderate the relationship between personality traits and burnout, particularly enhancing the protective effects of emotional stability and extraversion. These findings point to the necessity of integrating digital competence training into faculty development programs. Structured initiatives that target both technical skills and digital pedagogical strategies may not only improve instructional quality but also serve as a preventive measure against burnout in allied health education settings [22]. Given the limited but growing body of evidence, this remains a critical area for future investigation.

### 4.6. Theoretical Frameworks

The Job Demands–Resources model was the most commonly applied theoretical framework, followed by Maslach’s multidimensional theory of burnout, Conservation of Resources theory, and Self-Determination Theory (Figure 2). Some studies did not explicitly identify a theoretical framework. Research employing the JD-R model provided the most comprehensive examinations of interaction effects between different factors, conceptualizing leadership as an organizational resource, personality traits as personal resources, and digital competence as a technical resource that may buffer against job demands [14].

### 4.7. Methodological Quality

Quality assessment using the MMAT revealed considerable variation in methodological rigor. Common limitations included non-representative sampling (especially reliance on convenience samples), cross-sectional designs limiting causal inference, low response rates raising selection bias concerns, and inconsistent control for potential confounding variables. Qualitative studies generally demonstrated stronger methodological quality compared to quantitative approaches, with clear reporting of data collection and analysis procedures.

### 4.8. Summary of Research Questions and Key Findings

To explicitly address the defined research questions (RQs), the key findings of this scoping review are summarized below:

**RQ-1.** 
*What is the current state of research on the relationship between leadership approaches and faculty burnout in allied healthcare education?*


The literature consistently shows that transformational leadership is associated with reduced faculty burnout. Leadership behaviors that promote autonomy, recognition, and support serve as protective factors. In contrast, laissez-faire and overly directive styles are linked with increased burnout.

**RQ-2.** 
*How does the existing literature characterize the relationship between personality traits and burnout vulnerability among allied healthcare educators?*


Personality traits such as emotional stability, extraversion, conscientiousness, and agreeableness demonstrate protective associations against various dimensions of burnout. These traits appear to moderate the educator’s resilience to work-related stressors.

**RQ-3.** 
*To what extent has digital competence been examined as a potential moderating factor in burnout among allied healthcare educators?*


Although research on this topic is still emerging, digital competence is increasingly recognized as a key job resource. Higher digital skills are associated with lower emotional exhaustion and greater job satisfaction, especially in post-pandemic hybrid and online teaching environments.

**RQ-4.** 
*What are the significant gaps in the current literature regarding leadership, personality, digital competence, and burnout in allied healthcare education contexts?*


Key gaps include:
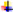
A lack of research in non-Western cultural contexts.
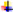
A predominance of cross-sectional studies, limiting causal inference.
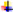
Limited exploration of interaction effects between leadership, personality, and digital competence.
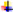
A lack of intervention studies and discipline-specific analysis outside of nursing education.

These findings provide an integrated understanding of the individual and contextual contributors to faculty burnout in allied healthcare education and suggest clear directions for both research and institutional policy.

## 5. Discussion

To offer a nuanced understanding of burnout in allied healthcare education, the discussion is organized around four key subsections: state of the literature, leadership implications, personality traits, and digital competence. These dimensions were chosen for in-depth analysis based on their recurring presence and significance across the included studies, as well as their relevance in guiding institutional policy and individual strategies.

### 5.1. State of the Literature

This scoping review reveals both established knowledge patterns and significant gaps in the current literature regarding leadership, personality, digital competence, and burnout in allied healthcare education. The field demonstrates increasing research interest, particularly since 2020, likely reflecting heightened concerns about educator wellbeing during the COVID-19 pandemic. However, several important limitations characterize the current state of knowledge.

First, geographical representation remains heavily skewed toward Western contexts, limiting understanding of cultural variations in burnout dynamics [4]. Second, most studies employ cross-sectional designs, constraining causal inference regarding relationships between variables. Third, interactive effects between different factors (leadership, personality, and digital competence) remain underexplored, despite theoretical frameworks suggesting important interaction possibilities.

Nevertheless, the available literature provides valuable insights into factors associated with burnout vulnerability and resilience among allied healthcare educators. The consistency of certain findings across studies—particularly regarding transformational leadership benefits and personality traits’ protective effects—suggests promising directions for intervention development.

### 5.2. Leadership Implications

The literature consistently identifies transformational leadership approaches as protective against faculty burnout. This leadership style’s emphasis on inspiration, individualized consideration, intellectual stimulation, and idealized influence appears particularly well-suited to professional educational contexts were faculty autonomy and meaningful work matter greatly [31].

Qualitative studies provide practical insights into specific leadership behaviors valued by faculty. Distributed leadership approaches that enhance faculty autonomy appear particularly valuable in professional education settings, where faculty members bring substantial expertise and value professional agency. This aligns with self-determination theory perspectives emphasizing autonomy as a core psychological need [24].

Recognition aligned with individual values represents another actionable leadership strategy with minimal resource requirements. The differentiated approach to recognition described across several studies acknowledges that faculty members may derive satisfaction from diverse aspects of their professional roles, requiring leaders to know their team members as individuals rather than applying generic appreciation strategies [23].

Resource advocacy emerges as a critical leadership function in resource-constrained settings. When institutional resources are limited, department chairs and program directors who effectively advocate for their faculty—securing technology, administrative support, or professional development opportunities—create protective buffers against burnout.

The emphasis on community cultivation highlights the importance of social resources in burnout prevention. Leaders who intentionally foster collegial communities create environments where informal support networks can flourish, enhancing resilience during challenging periods. This aligns with conservation of resources theory, which emphasizes the value of social resources in sustaining wellbeing under stress [33].

### 5.3. Personality and Burnout

The literature confirms that personality traits represent significant intrapersonal factors in burnout vulnerability among allied healthcare educators. The protective effects of extraversion, agreeableness, conscientiousness, emotional stability, and openness to experience against burnout dimensions align across studies in different contexts, suggesting consistency in these relationships.

The mechanisms through which personality influences burnout likely operate through multiple pathways. Extraverted individuals may maintain broader social support networks and engage in more positive interactions with students and colleagues, enhancing their access to social resources [25]. Agreeable individuals may experience more positive workplace relationships and interpret challenging interactions more charitably [26].

Conscientious educators may employ more effective work organization strategies, maintaining boundaries that prevent work overload while deriving satisfaction from goal achievement [28]. Emotionally stable individuals demonstrate greater resilience to stressors through more balanced appraisals and adaptive coping mechanisms [34]. Open individuals may approach challenges with greater flexibility and interpret difficulties as growth opportunities rather than threats [27].

These findings suggest that personality assessment may have value in identifying educators at heightened burnout risk, enabling targeted support interventions. However, it is essential to avoid deterministic interpretations; personality represents a risk or protective factor rather than destiny. As Angelini (2023) notes in a recent systematic review, contextual factors significantly moderate the expression of personality–burnout relationships [20].

### 5.4. Digital Competence as a Resource

Although digital competence represents the least studied area among the reviewed concepts, emerging evidence supports conceptualizing it as a valuable job resource within the JD-R framework. The preliminary findings indicating moderating effects of digital competence on personality–burnout relationships suggest several potential mechanisms worth further exploration.

Digitally competent extraverts may leverage technology to maintain broader support networks and engage more effectively with students and colleagues, enhancing their access to social resources. Conscientious individuals with strong digital skills may achieve greater efficiency in administrative tasks, freeing time for more rewarding aspects of their roles. Emotionally stable educators with digital competence may experience less technostress when navigating digital teaching environments, maintaining their characteristic resilience [22]. These findings align with recent research suggesting that digital skills constitute increasingly essential competencies for educator effectiveness and wellbeing in contemporary educational environments [14]. The identification of digital competence as a potential moderating factor rather than simply a direct influence highlights the importance of considering interactive effects in burnout research, acknowledging that the same resource may operate differently depending on individual characteristics.

### 5.5. Cross-Cultural Considerations

The limited representation of non-Western contexts in the current literature represents a significant gap, particularly given evidence of cultural variations in burnout expressions, leadership expectations, and personality manifestations. The few studies conducted in non-Western settings suggest both similarities and important differences in how burnout dynamics operate across cultural contexts [19].

In collectivist cultures, for example, leadership approaches emphasizing group harmony and hierarchical respect may function differently than in more individualistic Western settings. Similarly, the relative importance of different personality traits in predicting burnout may vary across cultural contexts, with potentially different mechanisms linking personality to burnout outcomes.

This gap highlights the need for greater cultural diversity in burnout research, as called for by several authors [4,5]. Future research should explicitly examine cultural factors as potential moderators of the relationships between leadership, personality, digital competence, and burnout among allied healthcare educators.

## 6. Research Gaps and Future Directions

This scoping review identified several significant gaps in the current literature that warrant attention in future research:

***Limited cultural diversity:*** Future studies should examine burnout dynamics across diverse cultural contexts, explicitly investigating how cultural factors moderate relationships between key variables.***Predominance of cross-sectional designs:*** Longitudinal studies are needed to establish directional relationships and examine how burnout, leadership perceptions, and digital competence evolve over time.***Insufficient examination of interaction effects:*** While the JD-R model suggests important interactions between different types of resources and demands, few studies have explicitly tested these interaction hypotheses. Future research should examine how leadership, personality, and digital competence interactively influence burnout outcomes.***Limited intervention research:*** Despite substantial descriptive literature, few studies have tested interventions targeting leadership development, personality-informed support, or digital competence enhancement as burnout prevention strategies. Intervention studies with rigorous experimental designs would significantly advance the field [35].***Inadequate consideration of discipline-specific factors:*** While nursing education is well-represented, some allied health disciplines receive minimal attention. Future research should examine discipline-specific challenges and resources that may influence burnout dynamics.***Measurement inconsistency:*** Greater standardization in measurement approaches would facilitate cross-study comparisons and meta-analytic integration of findings.***Digital competence as an emerging area:*** Given the accelerating digitalization of healthcare education, further research specifically examining digital competence as both a potential stressor and resource deserves priority attention [8].

## 7. Implications for Practice

Based on the synthesized literature, several practical implications emerge for addressing burnout among allied healthcare educators:

### 7.1. Institutional Policy Recommendations

Leadership development programs specifically addressing burnout prevention strategies, including recognition approaches, resource advocacy skills, community cultivation, and workload management techniques.Workload policies that explicitly acknowledge all dimensions of faculty responsibility (teaching, clinical supervision, research, service) with transparent allocation formulas and protection against continuous overload.Recognition systems that celebrate diverse contributions aligned with institutional mission while allowing for individualization based on faculty values and career stage.Digital competence development initiatives that provide both technical training and pedagogical application support, recognizing digital skills as critical professional competencies rather than optional enhancements.Community-building structures including formal mentoring programs, collaborative teaching opportunities, and interdisciplinary connections to enhance social resource availability.

### 7.2. Individual Strategies for Educators

Self-awareness development regarding personality tendencies and their potential implications for burnout vulnerability, enabling proactive implementation of personalized protective strategies.Strategic resource cultivation aligned with individual needs and preferences, including professional networks, mentoring relationships, and skill development priorities.Digital competence enhancement through targeted professional development, peer learning communities, and incremental implementation of digital approaches aligned with teaching values.Boundary management practices appropriate to individual work styles and personality tendencies, acknowledging that effective boundaries may look different across personality profiles.Leadership influence strategies that effectively communicate needs and advocate for necessary resources within institutional constraints, recognizing that upward influence represents an important professional skill.

For instance, digital competence programs might include institution-sponsored workshops on learning management systems (e.g., Moodle, Blackboard), video-based teaching tools (e.g., Panopto, Zoom), and data privacy essentials, tailored to varying faculty proficiency levels. Similarly, leadership training could incorporate modules on empathetic communication, workload distribution, and burnout recognition—using case-based simulations to help department heads and program directors respond proactively to early signs of educator fatigue.

## 8. Limitations

This scoping review has several limitations that should be considered when interpreting its findings. First, despite comprehensive search strategies, some relevant studies may have been missed, particularly those published in non-English languages or in non-indexed journals. Second, the broad scope of the review, encompassing multiple concepts and disciplines, necessitated somewhat general synthesis rather than deep examination of any single factor. Third, the heterogeneity of included studies in terms of methodological approaches, measurement instruments, and contextual factors limited direct comparisons across studies.

In addition, many of the included studies relied on cross-sectional designs and convenience sampling, which introduce significant methodological weaknesses. These limitations constrain the ability to draw causal inferences and raise concerns about representativeness and generalizability. While associations between variables such as leadership, personality, and burnout are frequently reported, the predominance of these weaker designs means that findings should be interpreted as correlational rather than causal. Finally, the quality assessment revealed methodological shortcomings across several included studies, including low response rates and limited control for confounding variables, further underscoring the need for caution in interpreting the evidence base.

An additional methodological consideration concerns the inclusion of both primary empirical studies and theoretical/review papers. While we included selected theoretical frameworks [10,24] and reviews [8,19,22] to provide conceptual grounding and contextualize findings, we acknowledge the potential for amplification bias when synthesizing findings from both primary studies and secondary sources. To mitigate this concern, we:

Clearly distinguished between primary empirical studies (n = 10, 67%), systematic reviews/meta-analyses (n = 2, 13%), theoretical papers (n = 2, 13%), and general reviews (n = 1, 7%) in Table 1 through explicit “Study Type” classification. Attributed findings primarily to primary empirical evidence throughout the synthesis, using review papers for theoretical context and convergent validity rather than as independent empirical weight. Conducted sensitivity checks confirming that core findings remained robust when based solely on primary empirical studies.

Cross-referencing revealed minimal overlap between primary studies in our review and those included in the systematic reviews we cited. For instance, Angelini’s (2023) review focused broadly on healthcare professionals across various settings, not specifically allied healthcare educators [19] and our primary studies [14,18,21] either post-dated Angelini’s search or focused on specific contexts (COVID-19, digital competence) not covered in that review. Similarly, reviews on digital competence examined general educational contexts with limited overlap with our allied healthcare education focus [8,22].

Despite these transparency measures, readers should recognize that the evidence base for some assertions (particularly regarding personality-burnout relationships) draws from both primary allied healthcare education studies and broader healthcare systematic reviews, which may amplify the apparent strength of certain findings. Future research would benefit from additional primary investigations specifically in allied healthcare education settings to confirm whether patterns observed in broader healthcare populations fully generalize to this specific context.

## 9. Conclusions

This scoping review mapped the current state of knowledge regarding leadership approaches, personality traits, digital competence, and burnout among allied healthcare educators. The findings confirm the important role of transformational leadership in reducing burnout risk, highlight the protective capacity of certain personality traits against burnout dimensions, and provide preliminary evidence for digital competence as a valuable job resource in increasingly digital educational environments.

However, these findings should be interpreted with caution, as the evidence base is predominantly derived from Western contexts, limiting the extent to which the conclusions can be generalized globally. Significant research gaps remain, particularly regarding cultural diversity, interaction effects between different factors, and intervention effectiveness. Future research addressing these gaps would substantially advance understanding of burnout dynamics in allied healthcare education and inform more effective prevention and mitigation strategies.

By identifying both individual and contextual factors influencing burnout among allied healthcare educators, this review provides an empirical foundation for multilevel interventions addressing this significant challenge. Effective approaches will likely combine institutional policy changes, leadership development initiatives, and individual support strategies tailored to educator characteristics and needs.

As allied healthcare education continues evolving to meet healthcare system demands, sustaining the wellbeing of educators represents a critical priority—not only for the individuals themselves but also for educational quality, student experiences, and ultimately, healthcare workforce development.

## Figures and Tables

**Figure 1 healthcare-13-02810-f001:**
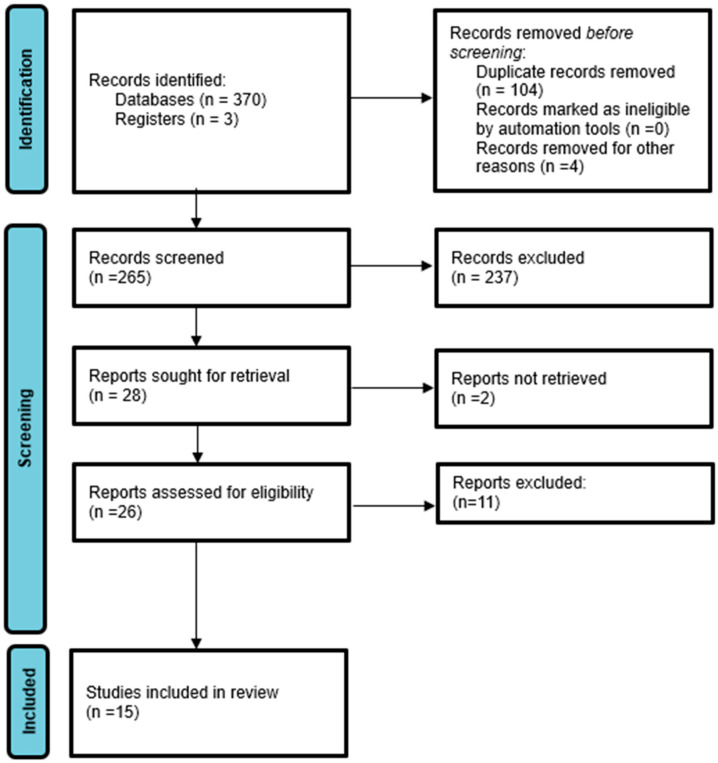
PRISMA 2020 flow diagram.

**Figure 2 healthcare-13-02810-f002:**
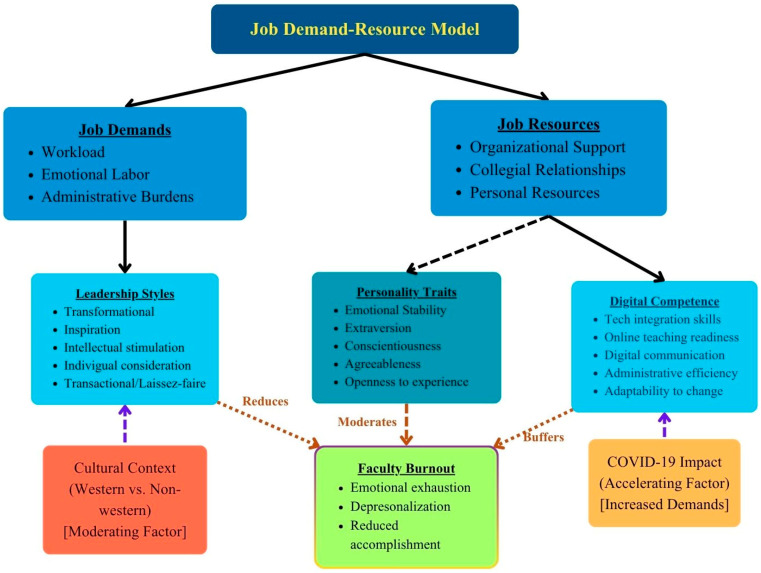
Conceptual framework model. Note: Dashed line arrows indicate an indirect or moderating relationship—specifically, that job resources interact with personality traits and digital competence to buffer against burnout, rather than exerting a direct effect. Solid arrows represent direct relationships.

**Table 1 healthcare-13-02810-t001:** Study Characteristics of Included Research on Allied Healthcare Educator Burnout.

S. No	Author, Year, and Origin	Study Type	Method	Discipline	Sample Size (n)/Response Rate (%)	Participant Characteristics
1	Garner et al., 2022. [18]	Primary Empirical	Survey and interviews	Medical education	Survey: n = 330 (76% response rate); Interviews: n = 12	Academic medical faculty during COVID-19 pandemic
2	Skogstad et al., 2007. [12]	Primary Empirical	Survey	Organizational psychology	4500/57%	Examined laissez-faire leadership and workplace outcomes
3	Chen et al., 2015. [19]	Primary Empirical	Mixed methods	Healthcare education	Theoretical paper; no empirical sample	Faculty from non-Western cultural contexts
4	Angelini, 2023. [20]	Systematic Review	Literature synthesis	Healthcare faculty	N/A *	Review of Big Five personality traits and burnout
5	Scherer et al., 2021. [21]	Primary Empirical	Survey	Higher education	739/Not available	Examined digital competence and teaching readiness
6	López-Meneses et al., 2020. [22]	Review	Literature synthesis	Educational context	N/A *	Examined digital competence as a moderating factor
7	Bakker et al., 2014. [10]	Theoretical/Conceptual	Literature review	Organizational psychology	N/A *	Examined Job Demands-Resources model
8	Gillespie et al., 2017. [23]	Primary Empirical	Program evaluation	Nursing education	n = 9 nursing faculty across 5 campuses	Nursing faculty and recognition approaches
9	Ryan & Deci, 2000. [24]	Theoretical/Conceptual	Literature review	Psychology	N/A *	Self-determination theory
10	De la Fuente-Solana et al., 2021. [25]	Primary Empirical	Survey	Nursing	95/Not specified	Examined burnout in specialized nursing context
11	Dionigi et al., 2020. [26]	Primary Empirical	Survey	Healthcare volunteers	160/Not specified	Examined personality factors and 13 psychological health
12	S. W. Kim et al., 2019. [27]	Meta-analysis	Statistical synthesis	Educational research	k = 49 studies (38 countries); N = 2,828,216	Examined socioeconomic factors in academic outcomes
13	König et al., 2020. [14]	Primary Empirical	Survey	Teacher education	165/54	Digital competence during COVID-19 transition
14	Triyono et al., 2023. [8]	Bibliometric Analysis	Literature review	Vocational education	N/A *	Reviewed digital competence literature
15	Zellars et al., 2004. [28]	Primary Empirical	Survey	Organizational psychology	296/23	Examined personality traits and burnout

* Note: N/A indicates that the information was not applicable or not reported in the original study (e.g., no empirical sample, theoretical or review paper).

## Data Availability

This study is a scoping review and does not involve the collection of primary data. All data analyzed during this review are available within the published articles cited in the manuscript and can be accessed through the respective journal sources.

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
