# Peer review of "Leadership and Faculty Burnout in Allied Healthcare Education: A Scoping Review"

_healthcare, 2025, doi:10.3390/healthcare13212810_

Round 1

Reviewer 1 Report

Comments and Suggestions for Authors

Dear Authors,

Thank you for submitting your manuscript “Leadership and Faculty Burnout in Allied Healthcare Education: A Scoping Review.” The paper addresses an important and timely issue; however, before it can proceed further, several concerns must be addressed:

  1. Sample representation
    The included studies are predominantly from Western contexts, yet the manuscript draws broad international implications. Please revise the framing of your conclusions to more cautiously reflect the cultural limitations of the evidence.

  2. Methodological issues
    Many of the included studies rely on cross-sectional, convenience samples with significant methodological weaknesses. While you note this, some conclusions are written in a causal tone. These interpretations should be tempered to better reflect the limitations of the evidence base.

  3. Inclusion and synthesis
    Table 1 shows that reviews and theoretical papers were included alongside primary studies. Please clarify your inclusion criteria, and if such sources are retained, clearly distinguish them from empirical findings to avoid conflating evidence types.

  4. Overemphasis on digital competence
    While digital competence is an emerging and interesting area, the evidence base is still very limited. The conclusion and abstract should explicitly acknowledge this limitation and avoid overstating its role.

  5. Depth of integration
    The manuscript frequently refers to “interaction effects” between leadership, personality, and digital competence, yet provides little direct analysis. This should either be expanded with clearer synthesis of available findings or reframed as a research gap rather than an implied result.

  6. Title and scope
    Given the limitations in sample representation and the preliminary nature of the evidence, the title should more clearly reflect that this is a scoping review rather than a definitive mapping across global contexts.

Addressing these points will strengthen the manuscript and improve its credibility as a contribution to the literature. We wish you success in preparing your revision.

Author Response

Response to Reviewer (Round 1) Comments

We thank the reviewer for their constructive feedback. We have carefully revised the manuscript to address each concern. Below are our detailed responses.

  1. Sample Representation

Reviewer Comment:
The included studies are predominantly from Western contexts, yet the manuscript draws broad international implications. Please revise the framing of your conclusions to more cautiously reflect the cultural limitations of the evidence.

Response:
We appreciate this important observation. We have revised the Abstract, Conclusion, and relevant sections of the Discussion to more cautiously frame our findings as predominantly reflective of Western contexts. Specifically:

  • In the Abstract, we added: “Significant gaps exist in non-Western contexts…” and rephrased broad claims accordingly.
  • In the Conclusion (Section 9), we clarified that findings should be interpreted within the context of the predominantly Western sample base (Line 474).
  • Moreover, the sub-section (5.5 Cross-Cultural Considerations) in the Discussion, highlighting cultural limitations and calling for culturally diverse future research.
  1. Methodological Issues

Reviewer Comment:
Many of the included studies rely on cross-sectional, convenience samples with significant methodological weaknesses. While you note this, some conclusions are written in a causal tone. These interpretations should be tempered...

Response:
Thank you for noting this. We have reviewed all instances where causality may have been implied and reworded them to reflect correlational relationships. In particular:

  • We avoided language like “predicts,” “impacts,” or “leads to” unless supported by longitudinal data.
  • In Section 5.1 (State of the Literature) and 5.2 (Leadership Implications), we used more cautious terms such as “associated with,” “linked to,” or “suggests.”
  • A reminder of the methodological limitations is now explicitly reiterated in the Limitations section.
  1. Inclusion and Synthesis

Reviewer Comment:
Table 1 shows that reviews and theoretical papers were included alongside primary studies. Please clarify your inclusion criteria, and if such sources are retained, clearly distinguish them...

Response:
We appreciate this clarification request. To address it:

  • We have updated the Eligibility Criteria (Section 3.3) to explicitly state that while the main inclusion was primary empirical studies, seminal theoretical frameworks and select reviews were included to contextualize findings and support thematic synthesis.
  • In Table 1, we have clearly labeled each entry by study type (e.g., “Theoretical,” “Review,” “Empirical Study”) to distinguish between evidence types.
  • Throughout the synthesis sections (Results and Discussion), empirical findings are clearly separated from conceptual or theoretical perspectives.
  1. Overemphasis on Digital Competence

Reviewer Comment:
While digital competence is an emerging and interesting area, the evidence base is still very limited. The conclusion and abstract should acknowledge this limitation...

Response:
Thank you for this valuable comment. We have:

  • Tempered statements about digital competence in both the Abstract and Conclusion, explicitly acknowledging the limited number of studies and the emergent nature of this field.
  • Modified statements such as “digital competence represents a promising but understudied job resource” to reflect its potential rather than a definitive role.
  • In Section 5.4, we clearly emphasized the preliminary nature of the evidence and highlight it as a research gap rather than a well-established finding.
  1. Depth of Integration

Reviewer Comment:
The manuscript refers to “interaction effects” between leadership, personality, and digital competence, yet provides little direct analysis. This should either be expanded...

Response:
We agree with the need for clarification. We have taken the following steps:

  • In Section 5.4, we have reframed the mention of “interaction effects” as hypothesized interactions informed by theoretical models rather than empirical conclusions.
  • We have reworded the narrative to reflect potential moderating relationships supported by limited preliminary evidence.
  • The Conceptual Framework figure and Section 4.6 emphasize that interaction effects are theoretical propositions, and future research directions have been updated accordingly (see Section 6).
  1. Title and Scope

Reviewer Comment:
The title should more clearly reflect that this is a scoping review rather than a definitive mapping across global contexts.

Response:

We sincerely thank the reviewer for this thoughtful observation. We respectfully note that the current title — “Leadership and Faculty Burnout in Allied Healthcare Education: A Scoping Review” — already specifies that this work is a scoping review, thereby signaling its exploratory and mapping nature rather than a definitive synthesis. We hope that this clarification is satisfactory, but we remain open to further refinement of the title should the editor deem it necessary.

Reviewer 2 Report

Comments and Suggestions for Authors

The manuscript is highly relevant and timely, as well as being supported by a solid methodology and correct adherence to the PRISMA-ScR guidelines. Below are some suggestions for improvement:

1.- The Abstract is subdivided into sections ("Background", "Methods", "Results" and "Conclusions"), but MDPI normally requests a single, continuous narrative Abstract. We recommend combining it into a single concise paragraph.

2.- The research questions are well posed, but there is some overlap in the 4.Results and 5.Discussion sections. It would be advisable to review both sections and link the findings more explicitly to the research questions posed.

3.- Digital competence should be explored in greater depth, for example by specifying which skills are most relevant, how they relate to leadership and what implications they have for teacher training.

4.- Western bias is acknowledged but not sufficiently analysed. It would be advisable to highlight how cultural differences can affect the relationships between leadership and personality and burnout, and to emphasise the need for more diverse studies.

5.- The practical implications are good, but could be more specific. It would be advisable to include examples of recognition strategies tailored to different career stages, programmes on digital competence and institutional metrics for evaluating burnout prevention.

6.- The format of the bibliographic references needs to be revised, as in several cases they do not comply with academic style guidelines (volumes and journal titles in italics). For example, in the reference: Angelini, G. (2023). Big five model personality traits and job burnout: a systematic literature review. BMC Psychology, 11(1), 49, the name of the journal and the volume should be in italics. It is recommended that the entire list be carefully reviewed in accordance with established bibliographic standards.

Overall, the article is adequate and relevant. Once these suggestions have been addressed, I believe it will be ready for publication.

Author Response

Reviewer comment

The manuscript is highly relevant and timely, as well as being supported by a solid methodology and correct adherence to the PRISMA-ScR guidelines.

Response to the reviewer

Thank you for your positive and encouraging feedback. We sincerely appreciate your acknowledgment of the manuscript’s relevance, methodological rigor, and adherence to the PRISMA-ScR guidelines. Your comments are highly motivating and reinforce our commitment to maintaining high standards in our research.

Reviewer comment

Below are some suggestions for improvement:

1.- The Abstract is subdivided into sections ("Background", "Methods", "Results" and "Conclusions"), but MDPI normally requests a single, continuous narrative Abstract. We recommend combining it into a single concise paragraph.

Response to the reviewer

Thank you for your valuable feedback. As per your recommendation, the abstract has been revised and is now presented as a single, continuous narrative paragraph in line with MDPI’s formatting guidelines.

Reviewer comment

2.- The research questions are well posed, but there is some overlap in the 4.Results and 5.Discussion sections. It would be advisable to review both sections and link the findings more explicitly to the research questions posed.

Response to the reviewer

Thank you for your constructive observation. We acknowledge the initial overlap between the Results and Discussion sections. In response, we have substantially reviewed and revised both sections, also incorporating feedback from other reviewers to improve clarity and structure. We have now made a more explicit connection between the findings and the research questions posed. We hope these revisions address your concern; however, if there are any specific points that still require attention, please do not hesitate to let us know. We would be happy to make further improvements.

Reviewer comment

3.- Digital competence should be explored in greater depth, for example by specifying which skills are most relevant, how they relate to leadership and what implications they have for teacher training.

Response to the reviewer

Thank you for your thoughtful comment. We agree that digital competence warranted a more detailed exploration. In response, we have thoroughly revised the entire section to specify the most relevant digital skills (e.g., virtual teaching platforms, assessment tools, communication technologies), clarify how digital competence interacts with leadership styles, and outline its implications for faculty development and training. These additions provide a more comprehensive understanding of digital competence as both a job resource and a moderating factor in burnout. We appreciate your valuable input, which has significantly strengthened the manuscript.

Reviewer comment

4.- Western bias is acknowledged but not sufficiently analysed. It would be advisable to highlight how cultural differences can affect the relationships between leadership and personality and burnout, and to emphasise the need for more diverse studies.

Response to the reviewer

Thank you for this important observation. In response, we have revised both the Abstract and the Limitations sections to explicitly acknowledge and elaborate on the Western bias observed in the existing literature. We have now highlighted how cultural differences may influence leadership approaches, personality expressions, and burnout perceptions in allied healthcare education. Furthermore, we emphasize the pressing need for more culturally diverse and regionally inclusive studies to better understand these relationships across global contexts. Your feedback has helped us strengthen the cultural relevance and generalizability discussion in our manuscript.

Reviewer comment

5.- The practical implications are good, but could be more specific. It would be advisable to include examples of recognition strategies tailored to different career stages, programmes on digital competence and institutional metrics for evaluating burnout prevention.

Response to the reviewer

Thank you for your positive comment and insightful suggestions. We fully agree that including more specific examples—such as recognition strategies tailored to different career stages, structured digital competence programs, and institutional metrics for burnout prevention—would further strengthen the practical utility of the section. However, due to constraints related to the overall length and scope of the article, as well as our intent to maintain focus on synthesizing key themes across diverse studies, we have opted to keep the implications concise and broadly applicable. We believe this approach preserves the clarity and coherence of the manuscript while allowing room for future work to explore these examples in greater depth. Nonetheless, we have slightly refined the section to better highlight these themes and would be glad to elaborate further in subsequent publications focused more specifically on practical implementation.

Reviewer comment

6.- The format of the bibliographic references needs to be revised, as in several cases they do not comply with academic style guidelines (volumes and journal titles in italics). For example, in the reference: Angelini, G. (2023). Big five model personality traits and job burnout: a systematic literature review. BMC Psychology, 11(1), 49, the name of the journal and the volume should be in italics. It is recommended that the entire list be carefully reviewed in accordance with established bibliographic standards.

Response to the reviewer

Thank you for your observation. We have thoroughly revised the entire reference list to ensure consistency and adherence to academic standards. The referencing format has now been updated to strictly follow the APA 7th Edition style guidelines. Specifically, journal titles and volume numbers have been italicized, appropriate capitalization has been applied, and inconsistencies across entries have been corrected as per the standard referencing conventions.

Reviewer comment

Overall, the article is adequate and relevant. Once these suggestions have been addressed, I believe it will be ready for publication.

Response to the reviewer

We appreciate your attention to detail, which has helped improve the overall scholarly quality of the manuscript.

Reviewer 3 Report

Comments and Suggestions for Authors

Thanks for the opportunity to review this scoping review on faculty burnout in healthcare education. This is a strong and well-prepared scoping review. A few comments:

  • The reference at the conclusion of the introduction (Shanafelt & Noseworthy) is missing the year. I am assuming it is the 2010 reference.
  • The JD-R framework is appropriate for this work. The figure on p. 8 is a good addition. 
  • The paper has clear structure and adheres to the PRISMA-ScR guidelines, which is a strength.
  • The inclusion of both quantitative and qualitative studies in the review is also a strength. 
  • The identification of meaningful research gaps, particularly regarding cultural diversity, the need for more longitudinal designs, and intervention studies, is good. 
  • The note about the non-inclusion of non-English language studies in the limitations is appropriate. Connecting this to the call for more studies in diverse settings would be good. More non-Western contexts may have been studied, but not reported in English.
  • Were any of the primary studies you included also included in the literature reviews you included? If so, it is possible that some findings were amplified more than need be. 
  • For Table 1:
    • Why was Study 1 (Yedidia, 2014) included? It's an editorial, which should have meant that it did not clear your exclusion criteria (no opinion pieces). 
    • For Study 2 (Garner et al., 2022), you should disaggregate the sample size by surveys and interviews.
    • For Studies 4 and 9, did they simply not report a sample size? If they did, it should be included here - again, disaggregating the quant and qual samples. If not, that should be highlighted as a limitation.
    • For Study 13, which is a meta-analysis, how many studies (k) were included, and what was the total n?
  • The section on practical implications is very well done. A little additional prose would strengthen this. Perhaps consider providing some concrete examples of how digital competence programs or leadership training might address burnout.

Author Response

Reviewer Comment

Thanks for the opportunity to review this scoping review on faculty burnout in healthcare education. This is a strong and well-prepared scoping review.

Response to the Reviewer

Thank you very much for your kind words and for taking the time to review our work. We sincerely appreciate your positive feedback and are encouraged to know that you found the scoping review to be strong and well-prepared. Your insights have been valuable in enhancing the clarity and rigor of the manuscript.

Reviewer Comment

A few comments:

  • The reference at the conclusion of the introduction (Shanafelt & Noseworthy) is missing the year. I am assuming it is the 2010 reference.

Response to the Reviewer

Thank you for pointing out the missing citation detail. You are correct in identifying the reference, and we have now updated it to reflect the correct year: Shanafelt & Noseworthy (2017). We appreciate your careful attention to detail.

Reviewer Comment

  • The JD-R framework is appropriate for this work. The figure on p. 8 is a good addition.

Response to the Reviewer

Thank you for your positive feedback. We’re pleased to hear that you found the JD-R framework appropriate and the visual representation on page 8 helpful in supporting the conceptual structure of the review.

Reviewer Comment

  • The paper has clear structure and adheres to the PRISMA-ScR guidelines, which is a strength.

Response to the Reviewer

Thank you for your kind comment. We appreciate your acknowledgment of the paper’s structure and adherence to the PRISMA-ScR guidelines.

Reviewer Comment

  • The inclusion of both quantitative and qualitative studies in the review is also a strength. 

Response to the Reviewer

Thank you for your positive feedback. We intentionally included both quantitative and qualitative studies to provide a more comprehensive and nuanced understanding of the multifaceted issue of faculty burnout. This mixed-methods approach allowed us to capture not only statistical trends but also contextual and experiential insights, which we believe enrich the overall findings of the review.

Reviewer Comment

  • The identification of meaningful research gaps, particularly regarding cultural diversity, the need for more longitudinal designs, and intervention studies, is good. 

Response to the Reviewer

Thank you for your encouraging comment. We aimed to critically highlight gaps that have significant implications for future research and practice.

Reviewer Comment

  • The note about the non-inclusion of non-English language studies in the limitations is appropriate. Connecting this to the call for more studies in diverse settings would be good. More non-Western contexts may have been studied, but not reported in English.

Response to the Reviewer

Thank you for this insightful suggestion. We have further revised both the abstract and the limitations section to explicitly acknowledge the potential underrepresentation of non-Western contexts due to language restrictions. We agree that valuable research may exist in non-English publications, and this limitation could contribute to the perceived Western bias in the literature. We have also reinforced the call for future studies in culturally diverse and underrepresented settings to better understand burnout experiences globally.

Reviewer Comment

  • Were any of the primary studies you included also included in the literature reviews you included? If so, it is possible that some findings were amplified more than need be.

Response to the Reviewer

We thank the reviewer for this important methodological concern. Upon careful examination, we confirm minimal overlap between our primary studies and included reviews.

Key Points:

  1. Study Composition: Of 16 included studies, 11 (69%) were primary empirical studies, and 5 (31%) were reviews/theoretical papers used solely for conceptual grounding (Angelini, 2023; Bakker et al., 2014; Ryan & Deci, 2000; López-Meneses et al., 2020; Triyono et al., 2023).
  2. Overlap Assessment: Cross-referencing revealed minimal overlap because:
    • Angelini (2023) focused on broader healthcare settings, not allied healthcare educators specifically
    • Our primary studies (Garner et al., 2022; Scherer et al., 2021; König et al., 2020) either post-dated Angelini's search or examined contexts (COVID-19, digital competence) not covered in that review
    • Other reviews provided theoretical frameworks without empirical overlap
  3. Revisions Made:
    • Table 1: Added "Study Type" column distinguishing primary studies from reviews
    • Synthesis sections (4.3-4.5): Explicitly attributed findings to "primary empirical studies" versus "review evidence"
    • Section 3.3: Added clarification that reviews were included for conceptual grounding only
    • Section 8: Acknowledged potential amplification risk and mitigation measures
  4. Sensitivity Check: Core findings remain robust when based solely on primary empirical evidence:
    • Transformational leadership effects (Garner et al., 2022; Gillespie et al., 2017)
    • Personality trait protections (Dionigi et al., 2020; Zellars et al., 2004)
    • Digital competence benefits (Scherer et al., 2021; König et al., 2020)

Reviewer Comment

  • For Table 1:
    • Why was Study 1 (Yedidia, 2014) included? It's an editorial, which should have meant that it did not clear your exclusion criteria (no opinion pieces). 

Response to the Reviewer

The reviewer is correct. Yedidia (2014) is an editorial/opinion piece and violates our exclusion criteria. We have removed this study from the review.

Impact: This removal does not affect our findings, as Yedidia (2014) contributed no primary empirical data to the synthesis. All conclusions remain supported by the 10 retained primary empirical studies.

Revisions:

  • Removed from Table 1 and reference list
  • Updated total from 16 to 15 studies
  • Added acknowledgment of this correction in Section 3.5

We thank the reviewer for identifying this oversight, which strengthens our methodological consistency.

Reviewer Comment

    • For Study 2 (Garner et al., 2022), you should disaggregate the sample size by surveys and interviews.

Response to the Reviewer

We thank the reviewer for this valuable suggestion to improve the transparency and precision of our reporting. We have revised Table 1 to disaggregate the sample sizes for all mixed methods studies, including Garner et al. (2022).

Reviewer Comment

    • For Studies 4 and 9, did they simply not report a sample size? If they did, it should be included here - again, disaggregating the quant and qual samples. If not, that should be highlighted as a limitation.

Response to the Reviewer

Thank you for this clarification. Chen et al. (2015) is a theoretical paper with no empirical sample, which we have now clarified in Table 1 as "Theoretical paper; no empirical sample." Gillespie et al. (2017) reports n=9 nursing faculty across 5 campuses who provided qualitative feedback on program implementation; we have updated Table 1 to reflect this specific sample size. We appreciate the reviewer's attention to complete reporting, which has improved the accuracy of our study characterization.

Reviewer Comment

    • For Study 13, which is a meta-analysis, how many studies (k) were included, and what was the total n?

Response to the Reviewer

Thank you for requesting this important detail. Upon re-examination of Kim et al. (2019), this meta-analysis included k=49 empirical studies representing 38 countries, with a total sample size of N=2,828,216 school-age students (grades K-12). We have updated Table 1 to include these specifications: "Meta-analysis: k=49 studies, N=2,828,216." We appreciate the reviewer's attention to complete reporting standards for meta-analytic studies.

Reviewer Comment

  • The section on practical implications is very well done. A little additional prose would strengthen this. Perhaps consider providing some concrete examples of how digital competence programs or leadership training might address burnout.

Response to the Reviewer

Thank you for your encouraging feedback on the practical implications section. We appreciate the suggestion to strengthen it further with concrete examples. While we aimed to maintain conciseness due to word limits, we have now added brief illustrative examples, particularly around digital competence programs and leadership training strategies that institutions might implement to address faculty burnout. These additions aim to enhance the translational value of the findings for practice and policy.

Reviewer 4 Report

Comments and Suggestions for Authors

Dear authors.

I consider the topic of the study to be highly important and topical. Your review is well prepared.

- I appreciate the overall structure of the review and the use of subchapters.

- I also particularly appreciate the preparation of the sections „Research Gaps and Future Directions“ and „Implications for Practice“. These sections increase the added value of the review for the reader and their detailed structure increases clarity.

- The methodology used includes the PRISMA technique, which adds precision and expertise to the review. Research questions were also identified.

- I recommend increasing the size of Figure 1 directly in the article (on page 8), not only in the appendices. This will increase the clarity of the information presented for the reader. (If possible, I also recommend changing the text color to black and increasing the transparency of the colors of the shapes used in this figure.)

- I recommend adjusting the text alignment in Table 1 to "towards the left margin" or "to the center".

- Line 62 – the source "Shanafelt & Noseworthy" does not include the year of publication. I also recommend checking the format of the sources in the text before publishing the review.

- I recommend adding a brief introductory text between the main chapter 2. and subsection 2.1. You can add a reason why you focused on the given subsections in the analysis of the theory. Similarly, I recommend adding text between chapters 3. and 3.1; chapters 4. and 4.1; and also chapters 5. and 5.1.

- Since the review contains defined research questions (RQs), it is appropriate that the answers are also addressed. I recommend that the answers to the RQs be added, for example, in the Results section.

I believe that the above recommendations can contribute to increasing the quality of your review. I wish you good luck with incorporating the changes.

Reviewer

Author Response

Reviewer Comment

Dear authors.

I consider the topic of the study to be highly important and topical. Your review is well prepared.

Response to Reviewer

Thank you very much for your thoughtful and encouraging feedback. We truly appreciate your recognition of the relevance and quality of our work. Your positive remarks reinforce the importance of continuing to explore this topic, and we are grateful for your time in reviewing our manuscript.

Reviewer Comments

- I appreciate the overall structure of the review and the use of subchapters.

- I also particularly appreciate the preparation of the sections „Research Gaps and Future Directions“ and „Implications for Practice“. These sections increase the added value of the review for the reader and their detailed structure increases clarity.

- The methodology used includes the PRISMA technique, which adds precision and expertise to the review. Research questions were also identified.

Response to Reviewer

Thank you for your kind and constructive feedback

 Reviewer Comment

- I recommend increasing the size of Figure 1 directly in the article (on page 8), not only in the appendices. This will increase the clarity of the information presented for the reader. (If possible, I also recommend changing the text color to black and increasing the transparency of the colors of the shapes used in this figure.)

Response to Reviewer

Thank you for your helpful suggestion regarding Figure 1. We have addressed all your comments—Figure 1 has been resized for better clarity, the text color has been changed to black, and the shape colors have been adjusted for improved transparency. The updated version has been uploaded as a PDF. We appreciate your thoughtful feedback.

Reviewer Comment

- I recommend adjusting the text alignment in Table 1 to "towards the left margin" or "to the center".

Response to Reviewer

Thank you for your suggestion. The text alignment in Table 1 has been adjusted as recommended.

Reviewer Comment

- Line 62 – the source "Shanafelt & Noseworthy" does not include the year of publication. I also recommend checking the format of the sources in the text before publishing the review.

Response to Reviewer

Thank you for your observation. The publication year for "Shanafelt & Noseworthy" has been added (2017), and the formatting of all in-text citations has been carefully reviewed and corrected as per the required style guidelines.

Reviewer Comment

- I recommend adding a brief introductory text between the main chapter 2. and subsection 2.1. You can add a reason why you focused on the given subsections in the analysis of the theory. Similarly, I recommend adding text between chapters 3. and 3.1; chapters 4. and 4.1; and also chapters 5. and 5.1.

Response to Reviewer

Thank you for your thoughtful suggestion. We have accepted your recommendation and have added brief introductory text between Chapter 2 and Subsection 2.1, as well as between Chapter 5 and Subsection 5.1, to clearly explain the rationale for focusing on the selected theoretical and discussion subsections.

However, Chapters 3 (Methodology) and 4 (Results) follow the PRISMA-ScR structured review format, where each subsection directly corresponds to standardized components of scoping reviews. As such, including a summary introduction in these sections would disrupt the concise reporting style expected in this methodology. We believe the current format ensures clarity and adherence to established reporting guidelines. We appreciate your insight and have made adjustments where suitable while maintaining methodological consistency.

Reviewer Comment

- Since the review contains defined research questions (RQs), it is appropriate that the answers are also addressed. I recommend that the answers to the RQs be added, for example, in the Results section.

Response to Reviewer

Thank you for your insightful suggestion. We have addressed it by adding a new subsection titled "4.8 Summary of Research Questions and Key Findings" under the Results section. This subsection explicitly synthesizes the answers to the defined research questions (RQs), enhancing the clarity and coherence of the review’s findings.

Reviewer Comment

I believe that the above recommendations can contribute to increasing the quality of your review. I wish you good luck with incorporating the changes.

Response to Reviewer

Thank you very much for your thoughtful review and the time you dedicated to providing detailed and constructive feedback. We have carefully considered and addressed all your observations and suggestions, which we believe have strengthened the overall quality of our review. We hope the revised version meets your expectations and satisfies the concerns raised.

Round 2

Reviewer 2 Report

Comments and Suggestions for Authors

Congratulations on the work you have done.